# Are Short-Stay Units Safe and Effective in the Treatment of Non-Variceal Upper Gastrointestinal Bleeding?

**DOI:** 10.3390/medicina59061021

**Published:** 2023-05-25

**Authors:** Marcello Candelli, Maria Lumare, Maria Elena Riccioni, Antonio Mestice, Veronica Ojetti, Giulia Pignataro, Giuseppe Merra, Andrea Piccioni, Maurizio Gabrielli, Antonio Gasbarrini, Francesco Franceschi

**Affiliations:** 1Emergency, Anesthesiological and Reanimation Sciencese Department, Fondazione Policlinico Universitario A. Gemelli—IRCCS of Rome, 00168 Rome, Italy; maria.lumare01@icatt.it (M.L.); antonio.mestice01@icatt.it (A.M.); veronica.ojetti@policlinicogemelli.it (V.O.); giulia.pignataro@policlinicogemelli.it (G.P.); andrea.piccioni@policlinicogemelli.it (A.P.); maurizio.gabrielli@policlinicogemelli.it (M.G.);; 2Medical and Abdominal Surgery and Endocrine-Metabolic Scienze, Fondazione Policlinico Universitario A. Gemelli—IRCCS of Rome, 00168 Rome, Italy; mariaelena.riccioni@policlinicogemelli.it (M.E.R.);; 3Biomedicine and Prevention Department, Section of Clinical Nutrition and Nutrigenomics, Facoltà di Medicina e Chirurgia, Università degli Studi di Roma Tor Vergata, 00133 Rome, Italy; giuseppe.merra@uniroma2.it

**Keywords:** gastrointestinal bleeding, short stay unit, emergency department, overcrowding, peptic ulcer, anticoagulant

## Abstract

*Introduction*: Emergency Department (ED) overcrowding is a health, political, and economic problem of concern worldwide. The causes of overcrowding are an aging population, an increase in chronic diseases, a lack of access to primary care, and a lack of resources in communities. Overcrowding has been associated with an increased risk of mortality. The establishment of a Short Stay Unit (SSU) for conditions that cannot be treated at home but require treatment and hospitalization for up to 72 h may be a solution. SSU can significantly reduce hospital length of stay (LOS) for certain conditions but does not appear to be useful for other diseases. Currently, there are no studies addressing the efficacy of SSU in the treatment of non-variceal upper gastrointestinal bleeding (NVUGIB). Our study aims to evaluate the efficacy of SSU in reducing the need for hospitalization, LOS, hospital readmission, and mortality in patients with NVUGIB compared with admission to the regular ward. *Materials and Methods*: This was a retrospective, single-center observational study. Medical records of patients presenting with NVUGIB to ED between 1 April 2021, and 30 September 2022, were analyzed. We included patients aged >18 years who presented to ED with acute upper gastrointestinal tract blood loss. The test population was divided into two groups: Patients admitted to a normal inpatient ward (control) and patients treated at SSU (intervention). Clinical and medical history data were collected for both groups. The hospital LOS was the primary outcome. Secondary outcomes were time to endoscopy, number of blood units transfused, readmission to the hospital at 30 days, and in-hospital mortality. *Results*: The analysis included 120 patients with a mean age of 70 years, 54% of whom were men. Sixty patients were admitted to SSU. Patients admitted to the medical ward had a higher mean age. The Glasgow-Blatchford score, used to assess bleeding risk, mortality, and hospital readmission were similar in the study groups. Multivariate analysis after adjustment for confounders found that the only factor independently associated with shorter LOS was admission to SSU (*p* < 0.0001). Admission to SSU was also independently and significantly associated with a shorter time to endoscopy (*p* < 0.001). The only other factor associated with a shorter time to EGDS was creatinine level (*p* = 0.05), while home treatment with PPI was associated with a longer time to endoscopy. LOS, time to endoscopy, number of patients requiring transfusion, and number of units of blood transfused were significantly lower in patients admitted to SSU than in the control group. *Conclusions*: The results of the study show that treatment of NVUGIB in SSU can significantly reduce the time required for endoscopy, the hospital LOS, and the number of transfused blood units without increasing mortality and hospital readmission. Treatment of NVUGIB at SSU may therefore help to reduce ED overcrowding but multicenter randomized controlled trials are needed to confirm these data

## 1. Introduction

Emergency department (ED) overcrowding is a worldwide problem of great concern to health care systems, policy makers, and the public. ED overcrowding is defined as a situation in which demand for ED services exceeds the capacity of ED, resulting in patients waiting longer for care, longer wait times, and even boarding of patients at ED. The causes of ED overcrowding are complex and multifactorial and can include factors related to patient demand, hospital resources, and system-level issues. Key factors contributing to ED overcrowding include increased patient demand, a shortage of hospital beds, lack of resources in the community, and delays in patient flow within the hospital. ED is often the first point of contact for patients with acute illnesses or injuries, and demand for ED services has steadily increased over the years. This increase in patient demand is caused by factors such as an aging population, an increase in chronic diseases, and a lack of access to primary care. The shortage of hospital beds can lead to longer waiting times at ED as patients wait for a bed to become available. This is especially true for patients who need to be admitted to the hospital. The lack of resources in the community, such as primary care clinics and mental health services, can lead to patients seeking care at ED rather than receiving it through channels that are more appropriate. Delays in patient flow within the hospital, such as delays in test results, consultation with specialists, or admission to the hospital, can also contribute to overcrowding [1]. The consequences of ED overcrowding can be severe, leading to delays in treatment, increased patient morbidity and mortality, and lower patient satisfaction. ED overcrowding can also lead to higher health care costs, as patients may require more complex and expensive treatment when delays in care occur. To address ED overcrowding, several solutions have been proposed around the world. These include initiatives to improve patient flow, such as fast-track pathways for patients with minor injuries or illnesses, and the use of clinical decision units to care for patients who do not require hospitalization. Interventions to reduce unnecessary visits ED have also been proposed, such as improving access to primary care and promoting patient education [2]. It is important to note that while these solutions have proven effective in some cases, there is no one-size-fits-all solution for ED crowding. Implementing these solutions may require a tailored approach that considers the specific needs of each health system and population. Short-stay units (SSUs) in ED have been suggested as a potential solution to reduce overcrowding and improve patient flow. SSUs are designated areas within the ED, where patients can be treated for up to 24–72 h before either being discharged or transferred to another hospital unit. Several studies suggest that SSUs can effectively reduce ED overcrowding and improve patient flow. For example, an Italian review found that the introduction of a SSU resulted in a significant reduction in ED length of stay, risk of in –hospital acquired infections, and boarding time [3]. Other studies found that SSUs could reduce length of stay (LOS) and mortality in patients with chronic obstructive pulmonary disease (COPD) and acute heart failure (AHF), but at the cost of higher readmission rates [4,5]. Even in elderly patients, the incidence of adverse events may be lower when admitted to SSU than to a medical ward, as shown by Strom C et al. in an observational study in Denmark [6]. However, a review of 10 studies highlighted that the quality and evidence of safety and efficacy of SSU are low and that further studies are needed to compare usual care and SSU to better understand the potential benefits and limitations [7]. SSUs have been proposed for several conditions. Upper gastrointestinal bleeding (UGIB) is a common emergency that often requires urgent investigation and treatment and is one of the possible indications for hospital admission to a SSU. Studies evaluating SSUs are often performed on all patients who have access to them and include patients with UGIB [8]. At now, there are no studies that address the safety and efficacy of SSUs in the management of UGIB. The aim of our study was to evaluate the efficacy of SSU in reducing the need for hospitalization, length of hospital stay, hospital readmission, number of blood units transfused and mortality in patients with non-variceal UGIB (NVUGIB) presenting to ED.

## 2. Materials and Methods

This is a retrospective, monocentric observational study conducted using electronic medical records (EMR) of 120 patients presenting with NVUGIB from 1 April 2022 to 30 September 2022 at ED of Fondazione Policlinico Agostino Gemelli Hospital—IRCCS of Rome. Patients with gastrointestinal bleeding come to our ED in a variety of ways. They may present to our emergency department on their own, or they may call the public emergency service and be transported by ambulance, and finally, they may be transferred from other lower level hospitals that do not have the appropriate resources to diagnose, treat, and manage the pathology affecting the patient (spoke centres). In our study, we only included patients who came to our hospital on their own or through emergency services, and not those who were referred by our spoke centres. At our hospital, patients who come to the emergency department with a diagnosis of suspected gastrointestinal bleeding and do not require intensive care are admitted to our SSU as their first choice. If beds are not available at SSU, patients are assigned to a gastroenterology or internal medicine department. The medical staff of our SSU consists of all the doctors of ED, who take turns to take care of the patients admitted there, and a chief physician with expertise in gastroenterology.

We enrolled 60 patients who were admitted to our SSU from April 2022 to June 2022, and 60 patients who were admitted to a medical ward from July 2022 to September 2022, when the SSU was closed. The study aims is to compare patient outcomes and resource utilization between two groups: those admitted to a regular medical ward (control group) and those treated at ED in a SSU (intervention group). The study population consists of adult patients (18 years or older) who present to ED with UGIB, defined as acute blood loss from the upper gastrointestinal tract (hematemesis or melena). Patients with chronic UGIB, variceal bleeding and those requiring immediate surgical intervention or admission to intensive care unit (ICU), palliative care and patients with an initial prognosis of less than 6 months of life were excluded. Electronic medical recorder data were collected for each patient, including demographic information, medical history, vital signs, laboratory and endoscopic results, treatment, therapies, disposition (admission or discharge), hospital length of stay (LOS), time to endoscopy, healthcare utilization (e.g., number of blood units transfused) and, Rockwood clinical frailty scale. Outcome measures: The primary outcome measure was hospital LOS (from triage registration to discharge). Secondary outcomes include time to endoscopy (from triage registration to endoscopy), number of blood units consumed, hospital readmission at 30 days, and in-hospital mortality. Statistical analysis: Descriptive statistics has been used to summarize the patient characteristics and outcome measures for each group. Continuous data were described as mean and standard deviation or median and interquartile range. Categorical data were described as percentages. Bivariate analysis was used to compare the characteristics and outcomes between the control and intervention groups (Chi square test or Fisher’s exact test for categorical data and Student’s T-Test or Mann-Whitney U test for continuous data). Multivariate regression analysis has been used to adjust for potential confounding factors (e.g., age, comorbidities, severity of illness) and to estimate the effect of SSUs on the primary and secondary outcomes. 

Ethics and approvals: This study has been conducted in accordance with ethical guidelines and regulations, and approved from the Ethical Board of Catholic University of the Sacred heart of Rome, Italy, (ID:5378)

## 3. Results

The current analysis includes 120 patients (mean age 69.6 ± 0.7 years), of whom 54% were men. Overall, 60 of the patients (50) were admitted to short stay unit (SSU) and 60 (50%) were admitted to an internal medicine ward. In Table 1 are showed patients demographic, clinical and laboratory and outcomes data. 

The main duration of symptom before admission was 34 ± 23 h with a wide range (2–72 h). However no difference in duration of symptoms before admission was found between patients admitted in the SSU and controls (32 ± 25 vs. 35 ± 22 h, *p* = 0.09). The distribution of symptoms duration before addmission was similar between groups (*p* = 0.68). Patients admitted directly to medical ward has a higher mean age (72.6 ± 16 vs. 66.4 ± 16 years; *p* = 0.03), a higher probability to have 2 or more comorbidities (55 vs. 35%; *p* = 0.03)), a higher chance to had a history of cerebrovasculare disease (13 vs. 2%; *p* = 0.03) and active cancer (30 vs. 7%; *p* = 0.03). No difference in use of anticoagulants or antiplatelets drugs was found between groups. The at-home use of proton pump inhibitors (PPI) was statistically lower in patients admitted to SSU then in controls (25 vs. 47%: *p* = 0.01). Among evaluated laboratory data only the international normalized ratio (INR) was found slightly but significantly higher in patients admitted to medical ward then in patients admitted to SSU (1.3 ± 1 vs. 1.1 ± 1; *p* = 0.04). The number of comorbidities between groups was showed in Figure 1. To evaluate the risk of bleeding in enrolled patients we used the Glasgow Blatchford bleeding Score (GBS). Study and control groups showed similar GBS). Both the median scores and the distribution of the Rockwood Fraility Clinical Score did not differ between case and control (*p* = 0.83 and 0.64, respectively). Finally, the number of patients with a RCFS > 4 was similar between patients admitted in the SSU and in the medical ward (Table 1).

Thirty-two patients had an history of previous gastrointestinal beeding, 13 in the SSU group and 19 in control group (22 vs. 32%; *p*: 0.22) In hospital mortality and readmisson to the hospital at 30 days were very low and similar between group. LOS, time to endoscopy, number of patients who need trasfusion, number of unit of blood transfused were significantly lower in patients admitted to SSU then in controls. Finally, we found no differences in mortality and the need for hospital readmission 30 days after discharge (Table 1). In Table 2 are showed the sources of UIGB. 

Other sources non described in the table were 3 gastric antral vascular ectasias (GAVE, 1 in SSU), 2 Dieulafoy’s lesion (1 in SSU) and 5 neoplasms (1 gastric malignancy in SSU and 2 in control group, 1 duodenal malignancy in control group and 1 biliary neoplasm infiltrating duodenum in SSU). In Table 3 are described the Forrest’s classification of peptic ulcers in our patients. No statistical difference was found between studied groups.

None of the patients participating in the study required urgent or elective surgery during their hospital stay.

No difference in the number and type of hemostatic techniques was found between the study groups. Mechanical hemostasis with endoscopic clips was the most commonly used means of controlling and treating the source of bleeding. Thermocoagulation (with argon plasma coagulation or heater bipolar probe), injection of diluted (1:10,000) epinephrine, and injection of fibrin glue were also used alone or in conjunction with mechanical hemostasis. Table 4 lists all hemostatic techniques used in the patients studied.

Finally, we performed two multivariate linear regression for the two continue outcomes we found related with SSU admission (LOS and time to endoscopy). We corrected for age, sex and all the variables that at univariate analyses had a p level of at least 0.1. The only factor that resulted independently associated to a reduced LOS were the SSU admission (*p* < 0.0001). The atrial fibrillation was associated to an increased LOS (*p* < 0.01). The SSU admission resulted independently and significantly associated to a shorter time to endoscopy (*p* < 0.001). The others only factor associated to a reduced time to EGDS was the creatinine levels (*p* = 0.05). At the contrary, the at-home treatment with PPI was associated to a longer time to endoscopy (*p* < 0.05). 

## 4. Discussion

SSUs are used worldwide to reduce ED waiting times, overcrowding, and hospital admissions. However, according to a recent meta-analysis, there is still inconclusive evidence of their efficacy and safety due to heterogeneity of outcomes, pathologies considered, and admission criteria to SSU [7]. For example, a recent study of patients with heart failure showed that there were no differences in safety and efficacy between patients discharged from SSU and patients discharged directly from ED [9]. In contrast, other studies have found that SSU can reduce LOS for patients with atrial fibrillation, chest pain, and syncope [10,11,12]. The extreme diversity of pathologies studied in a SSU is likely the main cause of the conflicting results in the literature. Our study focused on NVUGIB, a condition not previously treated in a SSU. NVUGIBs are an important cause of ED visits and result in a high number of admissions to internal medicine and gastroenterology departments [13]. We retrospectively compared outcomes of patients with NVUGIB treated in a SSU or in a medical ward of our tertiary teaching hospital. Bleeding severity was assessed by GBS and did not differ between the 2 groups. Patient frailty assessed with the Rockwood Clinical Frailty Scale and time of onset of symptoms on arrival at the emergency department did not differ between the two groups studied and therefore do not appear to be factors that could explain the observed differences in LOS. However, some variables evaluated such as, age and the number of concomitant diseases were higher in patients treated as inpatients than in those admitted at SSU. For this reason, we adjusted the results for these potential confounders and for variables that showed significant differences between groups in univariate analysis. Even after correction, patients treated in the SSU had significantly lower LOS than patients admitted to an internal medicine ward. The shorter time from triage to admission underscores that at least part of the overall reduction in length of stay is due to the faster bed turnover in SSU compared with medical wards. Regarding the shortening of the time from endoscopy to discharge by SSU compared to the medical department, we can hypothesize that physicians in the internal medicine departments are more likely to look at the patient from all sides and spend more time resolving other, non-acute problems of the patient. The availability of diagnostic tools and care providers at any point in the day may have facilitated timely decision-making in the SSU compared to hospitalized patients. In addition, the emergency doctor treating patients at the SSU may face increased pressure from the ED to expedite discharge. Finally, the reduction in time to endoscopy we observed for patients admitted to SSU is certainly another important factor in reducing overall LOS. The time to endoscopy determined in our study was high and certainly higher than the time to endoscopy recommended in the main international guidelines. However, time to endoscopy was calculated based on triage registration rather than visit to ED. Given the overcrowding in emergency departments, the time between triage registration and physician visit can be very long. In addition, urgent endoscopy, especially for patients seen at night or on holidays, is performed in the operating room reserved for emergencies and competes with other surgical procedures. Obviously, unstable surgical patients are prioritized over stabilized patients with suspected gastrointestinal bleeding. We believe that the shorter time to endoscopy for patients admitted to SSU is related to endoscopists’ perception of a requests from the ED, which includes SSU, as more urgent than requests for in-hospital admitted patients. The shorter time to endoscopy is probably related to the lower number of transfusions in patients admitted to SSU. It is likely that earlier treatment of the bleeding source contributes to a lower need for red blood cell units. However, the lower hemoglobin levels in the control group are also a factor influencing this outcome. Creatinine is the only factor besides SSU admission associated with shortened time to endoscopy. Higher creatinine levels are associated with higher patient frailty and complexity, although the correlation with creatinine level remains after adjusting for these factors in this study. However, creatinine is an important risk factor for gastrointestinal bleeding and has been associated with increased mortality in several studies [14,15,16]. For this reason, the emergency physicians and endoscopists could be more motivated to request and perform early EGDS in these patients. In addition, the increase in creatinine levels in patients taking anticoagulants and antiplatelet agents could increase the concentrations of these agents and contribute to greater bleeding and clinical severity explaining the shorter time to endoscopy. In contrast, taking PPIs at home, as opposed to creatinine, is a factor that “reassures” physicians and reduces the extent of bleeding, resulting in a longer time to endoscopy. Another finding of our analysis concerns the INR value, which seems to be slightly but significantly higher in patients admitted to the internal medicine ward than in patients admitted to SSU. We also found that AF is associated with a significant increase in hospital LOS. These two findings are likely related. Some patients with AF are treated with vitamin K antagonists (anticoagulant medications) that increase INR; this means that gastrointestinal bleeding may be more important in these patients, require more time to ensure patients’ clinical stability, and have a higher risk of recurrence. Therefore, a longer LOS is required in patients with AF, regardless of the hospital unit to which they are admitted (SSU or medical department). Our study has limitations. It is a retrospective study, and any biases inherent in this model may be present. In particular, selection bias cannot be excluded. The study was conducted at a single center with extensive experience in the treatment of gastrointestinal pathologies, so generalization of the results is not possible. In addition, the study sample was designed to analyze differences in the primary outcome (LOS) rather than the other end-points, so the lack of a difference in mortality and rehospitalization at 30 days between groups may be due to a relatively small number of patients included. 

## 5. Conclusions

Management of NVUGIB in SSU allowed a significant reduction in time to endoscopy, length of hospital stay, and number of blood units transfused without increasing mortality and hospital readmission. Treatment of NVUGIB in SSU could help reduce overcrowding in ED. Multicenter randomized controlled trials are needed to confirm these results.

## Figures and Tables

**Figure 1 medicina-59-01021-f001:**
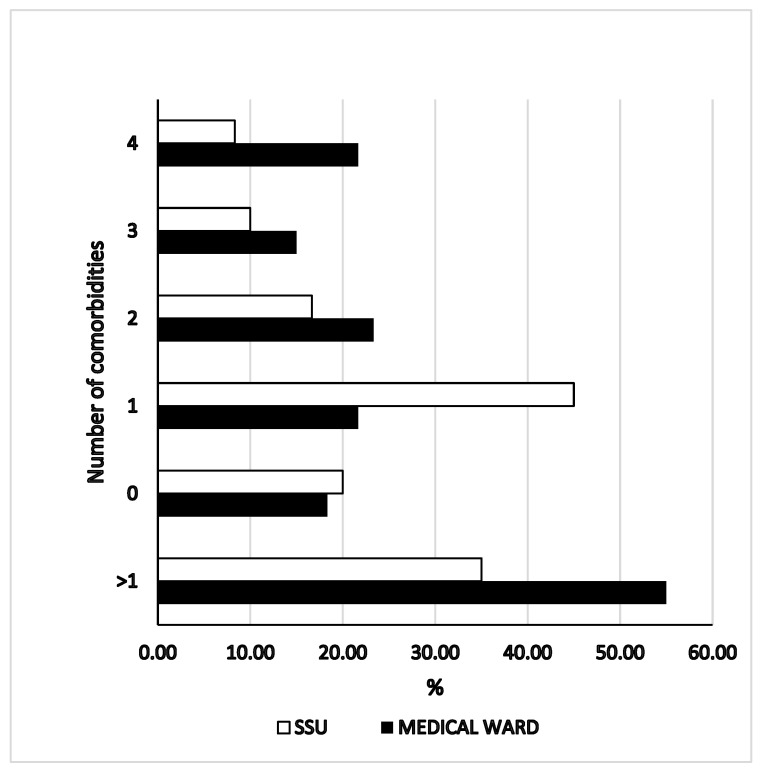
Number of comorbidities in patients admitted to medical ward or SSU. SSU: short stay unit. *p* = 0.03 between groups for 2 comorbidities or more.

**Table 1 medicina-59-01021-t001:** Patients’ demographic, comorbidities, laboratory, and outcomes data.

	All Patients (N = 120)	SSU (N = 60)	Medical Ward (N = 60)	*p*
Demographic data				
Age (years, M ± SD)	69.6 ± 16	66.4 ± 16	72.6 ± 16	**0.03**
Males N (%)	65 (54)	29 (48)	36 (60)	0.2
RCFS (Median and IQR)	4 (2–5)	4 (3–5)	4 (2–4)	0.83
Clinical presentation at ED				
Melena N (%)	110 (91.6)	55 (91.6)	55 (91.6)	1
Hematemesis N (%)	8 (6.7)	4 (6.7)	4 (6.7)	1
Rectal bleeding + UGIB N (%)	2 (1.7)	1 (1.7)	1 (1.7)	1
Duration of symptoms at admission (M ± SD)	34 ± 23	32 ± 25	35 ± 22	0.09
Comorbidities N (%)				
Diabetes	25 (21)	10 (17)	15 (25)	0.26
Hypertension	48 (40)	34 (57)	22 (37)	0.46
Coronary Heart Disease	21 (17)	9 (15)	12 (20)	0.52
Congestive Heart Failure	10 (8)	5 (8)	5 (8)	1
Chronic Liver Disease	6 (5)	2 (3)	4 (7)	0.7
Atrial Fibrillation	32 (27)	12 (20)	20 (33)	0.1
COPD	15 (12)	5 (8)	10 (17)	0.17
Active cancer	22 (18)	4 (7)	18 (30)	**0.002**
History of stroke	9 (7)	1 (2)	8 (13)	**0.03**
Chronic Kidney Disease	11 (9)	4 (7)	7 (12)	0.53
Previous bariatric surgery	6 (5)	4 (7)	2 (3)	0.7
Autoimmune diseases	6 (5)	3 (5)	3 (5)	1
VTE	6 (5)	3 (5)	3 (5)	1
N. of comorbidities >1	54 (45)	21 (35)	33 (55)	**0.03**
RCFS > 4	32 (27)	14 (23)	18 (30)	0.68
At-home treatment N (%)				
Anticoagulants	33 (27)	12 (20)	21 (35)	0.07
VKA	5 (4)	0 (0)	5 (4)	0.21
DOAC	22 (18)	10 (17)	12 (20)	0.64
Dabigatran	4 (3)	3 (5)	1 (2)	0.62
Apixaban	4 (3)	1 (2)	3 (5)	0.62
Rivaroxaban	4 (3)	0 (0)	4 (7)	0.36
Edoxaban	12 (10)	6 (10)	4 (7)	0.74
LMWH	5 (7)	1 (2)	4 (7)	0.36
Fondaparinux	1 (2)	1 (3)	0 (0)	0.81
Antiplatelets	29 (24)	13 (22)	16 (27)	0.52
ASA	23 (27)	11 (18)	12 (20)	0.82
Clopidogrel	10 (8)	5 (8)	5 (8)	1
Others antiplatelets	4 (3)	0 (0)	4 (7)	0.36
Dual antiplatelets therapy	8 ((7)	3 (5)	5 (8)	0.77
Anticoagulant + Antiplatelets	3 (2)	1 (2)	2 (3)	1
NSAIDs	12 (10)	5 (8)	7 (12)	0.54
Corticosteroids	8 (7)	2 (3)	6 (10)	0.27
PPI	33 (27)	15 (25)	28 (47)	**0.01**
Laboratory and vital signs (M ± DS)				
Hemoglobin (g/dL)	8.7 ± 2	9.0 ± 2	8.3 ± 2	0.11
WBC (×10^9^/L)	8794 ± 3350	8960 ± 3270	8629 ± 3447	0.62
Neutrophils (×10^7^/L)	7040 ± 6564	6450 ± 2900	7633 ± 8819	0.31
Plt (×10^9^/L)	260 ± 97	273 ± 104	247 ± 90	0.15
INR	1.2 ± 1	1.1 ± 1	1.3 ± 1	**0.04**
Na^+^ (mmol/L)	139 ± 3	139 ± 3	138 ± 4	0.43
K+ (mmol/L)	4.2 ± 1	4.3 ± 1	4.1 ± 1	0.15
Creatinine (mg/dL)	1.2 ± 1	1.1 ± 1	1.4 ± 1	0.06
SBP (mm/Hg)	110 ± 18	109 ± 16	112 ± 17	0.34
Heart Rate (beats/min)	94 ± 12	95 ± 14	93 ± 10	0.62
BUN (mmol/L)	9.6 ± 7	9.6 ± 6	9.7 ± 7	0.98
Glasgow-Blatchford Score *	10 (7–12)	10 (7–13)	10 (8–12)	0.89
Outcome N (%)				
Patients who need transfusion	69 (57)	28 (47)	41 (68)	**0.02**
Blood unit transfused *	1 (0–2)	0 (0–2)	2 (0–2)	**0.04**
Readmission at 30 days	3 (3)	1 (2)	2 (3)	1
In-hospital death	2 (2)	0 (0)	2 (3)	0.76
Admission to hospital	67 (56)	7 (12)	60 (100)	**<0.0001**
Outcome (M ± SD)				
Length of Hospital stay (h)	214 ± 209	126 ± 133	298 ± 212	**<0.0001**
Time to endoscopy (h)	66 ± 14	31 ± 39	104 ± 119	**<0.0001**
Time to admission (h)	45 ± 26	33 ± 21	67 ± 30	**<0.001**
Time from endoscopy to discharge	145 ± 181	99 ± 131	199 ± 207	**<0.0001**

Legend. SSU: short stay unit, M: media, SD: standard deviation, N: number, IQR: interquartile range, RCFS: Rackwood clinical fraility scale, ED: emergency deparment, COPD: chronic obstructive pulmunary disease, VTE: venous thromboembolism, VKA: vitamine K antagonist, DOAC: direct oral anticoagulants, LMWH: low molecular weight heparin, ASA: actylsalicylic acid, NSAIDS: non steroidal anti-inflammatory drugs, PPI: proton pump inhibitor, WBC: white blood cells, Plt: platelets, INR: international normalized ratio, Na^+^: sodium, K^+^: potassium, SBP: systolic blood pressure, BUN: Blood urea nitrogen, h: hours. * (Median and Interquartile Range). Significant p values are written in bold.

**Table 2 medicina-59-01021-t002:** Sources of gastrointestinal bleeding.

Type of NVUGIB N (%)	All Patients (120)	SSU (60)	Medical Ward (60)	*p*
Peptic Ulcer	66 (55)	42 (70)	24 (40)	**0.001**
Gastric Ulcer	35 (29)	23 (38)	12 (20)	**0.03**
Duodenal Ulcer	31 (26)	19 (32)	12 (20)	0.14
Erosive hemorrhagic gastritis	15 (12)	4 (7)	11 (18)	0.1
Angiodysplasia	8 (7)	3 (5)	5 (8)	0.72
Obscure gastrointestinal bleeding	10 (8)	4 (7)	6 (10)	0.51
Esophagitis	11 (9)	3 (5)	8 (13)	0.20
Other sources	10 (8)	4 (7)	6 (10)	0.74

Legend. NVUGIB: non-variceal upper gastrointestinal bleeding; N: number, SSU: short stay unit. Significant *p* values are written in bold

**Table 3 medicina-59-01021-t003:** Forrest’s classification in patients with peptic ulcer disease between groups.

Forrest Classification	All Patients (66)	SSU (42)	Medical Ward (24)	*p*
III	40 (61)	25 (60)	15 (62)	0.90
IIc	7 (11)	5 (12)	2 (8)	1.00
IIb	3 (4.5)	1 (2)	2 (8)	0.55
IIa	11 (17)	9 (21)	2 (8)	0.30
I	5 (7)	2 (5)	3 (12)	0.34

Legend. NVUGIB: non-variceal upper gastrointestinal bleeding; SSU: short stay unit.

**Table 4 medicina-59-01021-t004:** Endoscopic hemostatic treatment used.

Hemostasis N (%)	All Patients (120)	SSU (60)	Medical Ward (60)	*p*
Any techniques	35 (29)	17 (28)	18 (30)	0.84
Endoscopic clip	23 (19)	11 (18)	12 (20)	0.81
Epinephrine injection	9 (8)	5 (8)	4 (7)	0.99
Thermocoagulation	8 (7)	5 (8)	3 (5)	0.72
Fibrin glue	7 (7)	5 (8)	2 (3)	0.44
2 or more combined tool	s15 (13)	10 (17)	5 (8)	0.27

## Data Availability

Data are available upon reasonable requests to corresponding author.

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
