# Peer review of "Are Short-Stay Units Safe and Effective in the Treatment of Non-Variceal Upper Gastrointestinal Bleeding?"

_medicina, 2023, doi:10.3390/medicina59061021_

Round 1

Reviewer 1 Report

1. In describing the methods, the authors do not mention the principle of admission or selection of patients admitted to the emergency department and the SSU, which is probably important from a management point of view. Criteria?

2. The authors do not identify in the paper the proportion of patients in the SSU who had palliative care in relation to the ED - an important clinical issue also from the point of view of principles of care and planning.

3. The authors are invited to comment on the factors that influenced the reduction in LOS and ED after the introduction of SSU.

4. The authors should define the basic criteria and the professional level of the SSU according to experience and the organization of access to diagnostic and therapeutic services.

5. The authors should define the distribution of patients in the SSU according to surgical and non-surgical diseases and possible changes in the management algorithms.

6. What was the proportion of patients with fragility in the ED and SSU, and does this estimate the influence of the decision to admission? 

7. In the paper, the authors did not provide an important piece of information, i.e. the duration of clinical signs and symptoms before admission to the individual department and the fact whether this affects the decision.

8. How many patients were transferred from the SSU unit to clinical wards for further treatment or observation? What does this mean for consideration?

9. How do you justify different algorithms or decision criteria for the needs of emergency endoscopy in the ED and SSU?

10. Is there a relationship between creatinine and o-GF when talking about risk factors and liver function tests given that a large proportion of patients are on antiplatelet or anticoagulant therapy?

Author Response

We would like to thank the reviewer for the valuable constructive criticisms, which allowed us to improve the paper significantly

  1. In describing the methods, the authors do not mention the principle of admission or selection of patients admitted to the emergency department and the SSU, which is probably important from a management point of view. Criteria?

Patients with gastrointestinal bleeding come to our emergency department in a variety of ways. They may present to our emergency department on their own, or they may call the public emergency service and be transported by ambulance, and finally, they may be transferred from other lower level hospitals that do not have the appropriate resources to diagnose, treat, and manage the pathology affecting the patient (spoke centres). In our study, we only included patients who came to our hospital on their own or through emergency services, and not those who were referred by our spoke centres. At our hospital, patients who come to the emergency department with a diagnosis of suspected gastrointestinal bleeding and do not require intensive care are admitted to our SSU as their first choice. If beds are not available at SSU, patients are assigned to a gastroenterology or internal medicine department.

  1. The authors do not identify in the paper the proportion of patients in the SSU who had palliative care in relation to the ED - an important clinical issue also from the point of view of principles of care and planning.

We excluded all patients who required palliative care and all patients with an initial prognosis of less than 6 months of life. We added this information in the methods

  1. The authors are invited to comment on the factors that influenced the reduction in LOS and ED after the introduction of SSU.

The introduction of SSu in the ED has precisely the purpose of reducing the LOS of patients as it is a rapidly managed ward with a high turnover where patients are manged by the emergency physitians in dedicated places. For this reason they have an quick access to all diagnostics in the view of rapid management and discharge.

  1. The authors should define the basic criteria and the professional level of the SSU according to experience and the organization of access to diagnostic and therapeutic services.

The medical staff of SSU is composed of all the physicians of ED who take turns caring for the patients admitted there and a chief with expertise in gastroenterology.

  1. The authors should define the distribution of patients in the SSU according to surgical and non-surgical diseases and possible changes in the management algorithms.

None of the patients participating in the study required urgent or elective surgery during their hospital stay. We have added this statement to the results

  1. What was the proportion of patients with fragility in the ED and SSU, and does this estimate the influence of the decision to admission? 

All of our patients are evaluated at triage and assigned to level of frailty using the Rockwood Clinical Frailty Scale. We have included the data in the results.

  1. In the paper, the authors did not provide an important piece of information, i.e. the duration of clinical signs and symptoms before admission to the individual department and the fact whether this affects the decision.

We have added this information in the results and discussion

  1. How many patients were transferred from the SSU unit to clinical wards for further treatment or observation? What does this mean for consideration?

As described in the table 1, 7 patients had to be transferred from the SSU to a clinical ward for further treatment or observation

  1. How do you justify different algorithms or decision criteria for the needs of emergency endoscopy in the ED and SSU?

Like all hospitals, we do our best to follow international guidelines, but in the face of specific cases and in real life, the relationship between physician and endoscopist is certainly fundamental. We believe, as described in the discussion, that endoscopists, when contacted by emergency physicians and by ED, feel more pressure to perform an urgent examination than when called by a clinical ward physician.

  1. Is there a relationship between creatinine and o-GF when talking about risk factors and liver function tests given that a large proportion of patients are on antiplatelet or anticoagulant therapy?

Thanks for your suggestion. We have added it to the discussion

Reviewer 2 Report

Some aspects of the discussion are rather subjective, regarding the evaluation of the time elapsed to endoscopy. It looks, that in SSU the endoscopists are more active or the management is better organized! On the other hand  I do not understand the elapsed hours:  66 ± 14 for all, but 31 ± 39!! for SSU and 104 ± 119!! for Medical ward?  The newer ESGE guideline proposes up to 24 hours but not significantly longer time till the endoscopy!  The GBS is nor really different between the two groups, showing that the severity of the patients was not really different,  that also means, the the endoscopy should have been performed  much earlier, at around 24-36 hours in all cases?!    

Author Response

Thank you for the valuable comments that have enabled us to improve the paper considerably

Some aspects of the discussion are rather subjective, regarding the evaluation of the time elapsed to endoscopy. It looks, that in SSU the endoscopists are more active or the management is better organized! On the other hand  I do not understand the elapsed hours:  66 ± 14 for all, but 31 ± 39!! for SSU and 104 ± 119!! for Medical ward?  The newer ESGE guideline proposes up to 24 hours but not significantly longer time till the endoscopy!  The GBS is nor really different between the two groups, showing that the severity of the patients was not really different,  that also means, the the endoscopy should have been performed  much earlier, at around 24-36 hours in all cases?!    

We thank the reviewer for this consideration. Of course, like all hospitals, we try to keep up with international guidelines, but we also have to deal with real life issues. First, the time to perform endoscopy has been calculated based on the triage registration rather than the visit to ED. Given the overcrowding in emergency departments, the time between triage registration and physician visit is very long and can sometimes exceed 12 hours. In addition, urgent endoscopy, especially for patients seen at night or on holidays, is performed in the surgical room reserved for emergencies and competes with other surgical procedures. Obviously, priority is given to unstable surgical patients rather then stabilized suspected gastrointestinal bleeding. We added these consideration in the discussion.

Round 2

Reviewer 1 Report

The authors have made significant improvements to the paper and reworked the discussion on critical points in the paper.